# The Effectiveness of Probiotic *Lactobacillus rhamnosus* and *Lactobacillus casei* Strains in Children with Atopic Dermatitis and Cow’s Milk Protein Allergy: A Multicenter, Randomized, Double Blind, Placebo Controlled Study

**DOI:** 10.3390/nu13041169

**Published:** 2021-04-01

**Authors:** Bożena Cukrowska, Aldona Ceregra, Elżbieta Maciorkowska, Barbara Surowska, Maria Agnieszka Zegadło-Mylik, Ewa Konopka, Ilona Trojanowska, Magdalena Zakrzewska, Joanna Beata Bierła, Mateusz Zakrzewski, Ewelina Kanarek, Ilona Motyl

**Affiliations:** 1Department of Pathomorphology, the Children’s Memorial Health Institute, Aleja Dzieci Polskich 20, 04-730 Warsaw, Poland; e.konopka@ipczd.pl (E.K.); i.trojanowska@ipczd.pl (I.T.); j.bierla@ipczd.pl (J.B.B.); e.kanarek@ipczd.pl (E.K.); 2Outpatient Allergology and Dermatology Clinic, Patriotów St. 100, 04-844 Warsaw, Poland; aldona.ceregra@op.pl; 3Department of Developmental Age Medicine and Paediatric Nursing, Faculty of Health Sciences, Medical University of Bialystok, Szpitalna St. 37, 15-295 Białystok, Poland; emaciorkowska@o2.pl (E.M.); magdalena.maciorkowska@gmail.com (M.Z.); mateusz.zakrzewski4@wp.pl (M.Z.); 4Outpatient Allergology Clinic, the Children’s Memorial Health Institute, Aleja Dzieci Polskich 20, 04-730 Warsaw, Poland; b.surowska@ipczd.pl; 5Outpatient Dermatology Clinic, Chodakowska St. 8, 96-503 Sochaczew, Poland; azegadlo@op.pl; 6Department of Environmental Biotechnology, Lodz University of Technology, Wólczańska 171/173, 90-924 Łódź, Poland; ilona.motyl@p.lodz.pl

**Keywords:** probiotics, atopic dermatitis, food allergy, cow’s milk protein allergy, *Lactobacillus rhamnosus*, *Lactobacillus casei*

## Abstract

Probiotics seem to have promising effects in the prevention and treatment of allergic conditions including atopic dermatitis (AD) and food allergy. The purpose of this multicenter randomized placebo-controlled trial was to evaluate the effectiveness of a probiotic preparation comprising *Lactobacillus rhamnosus* ŁOCK 0900, *Lactobacillus rhamnosus* ŁOCK 0908, and *Lactobacillus casei* ŁOCK 0918 in children under 2 years of age with AD and a cow’s milk protein (CMP) allergy. The study enrolled 151 children, who—apart from being treated with a CMP elimination diet—were randomized to receive the probiotic preparation at a daily dose of 10^9^ bacteria or a placebo for three months, with a subsequent nine-month follow-up. The primary outcomes included changes in AD symptom severity assessed with the scoring AD (SCORAD) index and in the proportion of children with symptom improvement (a SCORAD score decreased by at least 30% in comparison with that at baseline). After the three-month intervention, both the probiotic and placebo groups showed a significant (*p* < 0.0001) decrease in SCORAD scores, which was maintained nine months later. The percentage of children who showed improvement was significantly higher in the probiotic than in the placebo group (odds ratio (OR) 2.56; 95% confidence interval (CI) 1.13–5.8; *p* = 0.012) after three months. Probiotics induced SCORAD improvement mainly in allergen sensitized patients (OR 6.03; 95% CI 1.85–19.67, *p* = 0.001), but this positive effect was not observed after nine months. The results showed that the mixture of probiotic ŁOCK strains offers benefits for children with AD and CMP allergy. Further research is necessary to assess the effect of probiotic supplementation on the development of immune tolerance (NCT04738565).

## 1. Introduction

Atopic dermatitis (AD), a chronic and recurrent skin disease of allergic origin that affects people with genetic predisposition, is characterized by intimal hyperplasia, dry skin, and periodic inflammatory and exudative lesions [1]. AD is characterized by epithelial barrier defects and dysregulation of both the innate and adaptive immune response against different triggers. Recently, cytokines and other mediators that play an important role in the pathogenesis of skin inflammation have become a target for new forms of therapy [2,3]. Drugs for which interleukin (IL)-4 and IL-13—the main cytokines of T helper 2 (Th2) immune activation—are the targets, are particularly represented. Dupilumab, a human anti-IL-4 receptor α monoclonal antibody that blocks IL-4- and IL-13-mediated signaling pathways, is the first biological drug approved by Food and Drug Administration for the treatment of moderate-to-severe AD in adolescents and adults [3]. Another approved therapeutic option for topical use is crisaborole, a phosphodiesterase-4 inhibitor which is a key regulator in the inflammatory cytokine cascade [3].

AD is one of the most common chronic childhood diseases, affecting approximately 10–20% of children in Europe [4]. AD usually starts in early childhood and may represent the first step of the so-called “atopic march”, which represents the natural history of allergic manifestations, characterized by a sequence of atopic diseases preceding the development of other allergic disorders later in life [5]. Typically, AD in infancy is followed by allergic rhinitis and/or asthma later in life [6]. Although the etiology of AD is still not clear, it has been reported that more than half of all children with AD are sensitized to one or more allergens with a predominance of food allergens [7,8], indicating their significant role in the early activation of the pro-allergic immune response.

Over the last few decades, there has been an increasing trend in the incidence of AD and food allergy, especially noticeable in highly developed countries, which is associated with a strict hygiene regime, increased use of detergents, low number of children per family, altered nutritional habits, frequent antibiotic therapy, low incidence of infectious diseases, and high numbers of Cesarean sections [9]. The Western lifestyle is believed to effect a change in the composition of gut microbiota which may activate pro-allergic mechanisms. Children with AD show a low biodiversity of their gut microbiota, particularly a lack of *Bacteroides* diversity, and a high prevalence of *Clostridium difficile* colonization [10,11]. Therefore, one of the prophylactic and therapeutic measures in allergies may involve the use of probiotics [12]. The World Allergy Organization guideline panel suggests to use probiotics for prevention of allergy in pregnant women at high risk for having an allergic child, in women who breastfeed infants at high risk of developing allergy, and in infants at high risk of developing allergy [13]. Probiotics are defined as live microorganisms which, when administered at the right dose, have a positive effect on human health [14]. Experts emphasize that the clinical effects of probiotics are strain-dependent. The probiotic strain *Lactobacillus rhamnosus* GG (LGG) is one of the few strains that had been evaluated in pediatric patients with AD in various research centers using the same study protocol [15,16,17,18]. The results of these studies show that it is not only strain selection but also the target population that matters. Beneficial therapeutic effects of LGG in AD were demonstrated in a population of Finnish children [15], whereas no such effects were found either in Dutch or German patients [16,17,18]. A similar outcome pattern was observed in studies evaluating LGG supplementation for the primary prevention of AD. The observed prevalence of AD after 2–4 and seven years was significantly lower when LGG was administrated to pregnant women and then to infants, but only in the Finnish population [19,20,21]. In contrast, in the German population such supplementation did not inhibit AD development. In fact, it increased the risk of wheezing by the age of two years [22]. The lack of probiotic effect in some populations may be a result of intestinal microbiota composition differences, which may be due to such factors as the geographic location. Thus, our research team attempted a search for new probiotic strains that could be used both in primary allergy prevention and as a complement to AD treatment. In 2009, three probiotic strains were selected out of 24 strains isolated from healthy Polish subjects [23]. We reported that the mixture of these strains had a synergistic effect on cytokine production in blood cell cultures obtained from AD infants [24]. The strains induced activation of Th1 cytokines and regulatory IL-10, and inhibition of pro-allergic IL-5.

In the present randomized double-blind placebo-controlled clinical trial we supplemented AD children under the age of two years with these selected probiotic strains in order to observe their effect on the clinical course of the disease assessed with the SCORing Atopic Dermatitis (SCORAD) index [25].

## 2. Materials and Methods

### 2.1. The Study Design

This was a randomized, double-blind, placebo-controlled, parallel-group study conducted at four Polish centers (the outpatient allergic clinic of the Children’s Memorial Health Institute in Warsaw, the outpatient allergic clinic of the Medical University in Białystok, the outpatient dermatology and allergic clinic in Warsaw, and the outpatient dermatology clinic in Sochaczew) between June 2012 and December 2015. The study had been approved by the Bioethical Committee of the Children’s Memorial Health Institute (decision number 4/KBE/2010). The subjects’ parents/guardians had been informed about the study objectives and design, and those who agreed to participate were required to provide a written consent prior to enrollment. The study was conducted in accordance with the ethical principles set out in the Declaration of Helsinki Guideline on Good Clinical Practice. The trial was registered in the ClinicalTrials.gov database (NCT04738565).

### 2.2. Patients

Subjects under the age of 2 years were enrolled in the study. Study inclusion criteria were diagnosis of AD according to Hanifin and Rajka’s criteria [26], SCORAD index > 10 points [25], suspected allergy to cow’s milk protein (CMP). Study exclusion criteria included age over 24 months, acute skin infections, the presence of other severe diseases, systemic corticosteroid treatment, and use of probiotics or antibiotics within the 6 weeks preceding study inclusion. The children who received antibiotic therapy during the study were also excluded.

All subjects met three out of Hanifin and Rajka’s four major diagnostic criteria for AD: onset in early childhood: chronic recurrent nature of lesions, pruritus, characteristic lesion morphology and distribution, and positive family history for atopy (allergic conditions in family members: parents and/or siblings) [26].

In order to confirm CMP allergy, first, cow’s milk was completely eliminated from the diet for three weeks. During this period, children received extensively hydrolyzed casein or whey-based formula, and nursing mothers were on dairy-free diets. A subsequent open cow’s milk challenge involved the administration of cow’s milk-based formulas to children (or dairy products to nursing mothers) for seven consecutive days. Those children who showed a diminished extent and severity of skin lesions during CMP elimination period and whose AD symptoms exacerbated again after cow’s milk was reintroduced were included in this study. The children with positive CMP challenge and an allergen-specific immunoglobulin (Ig) E concentration of ≥0.35 kUA/L in response to any allergen tested were considered to be allergen sensitized.

The children included in the study remained on CMP elimination diet throughout the study period (12 months) and received extensively hydrolyzed casein- or whey-based formulas; nursing mothers and weaned children remained on diets containing no CMP. Additionally, the food allergens triggering allergic responses were eliminated from the subjects’ diets. The parents/guardians were educated to ensure prevention of symptom exacerbations and the use of appropriate skin care methods that help restore the natural epidermal barrier, i.e., the use of emollients containing ceramides, unsaturated fatty acids, and cholesterol. Regular use of emollients was encouraged. Formulations containing urea were used to adequately moisturize the stratum corneum. The bath lasted up to 10–15 min and included the use of water at body temperature, no detergents, and the exclusive use of cleansing formulations, shampoos, and skin care products whose pH was close to 5.5, followed by the use of emollients within 15 min of gently drying the skin. The children with a severe course of AD additionally received oral antihistamines, topical antibiotic, and steroid ointments.

### 2.3. Probiotic Preparation

Study subjects received a mixture of three probiotic strains containing 1 billion (1 × 10^9^) colony-forming units (CFU) of selected bacteria in the following proportions: 50% of *Lactobacillus casei* ŁOCK 0919, 25% of *Lactobacillus rhamnosus* ŁOCK 0908, 25% of *Lactobacillus rhamnosus* ŁOCK 0900 (Latopic^®^, Biomed S.A., Cracow, Poland). The composition and anti-allergy properties of the study preparation had been patented (Republic of Poland patent license No. 212183 of 17 September 2007) and described in numerous literature reports [19,20,23]. The extent of our current knowledge of the ŁOCK strain genome resulted in the ŁOCK 0900 and ŁOCK 0908 species being reclassified from *Lactobacillus casei* to *Lactobacillus rhamnosus*, and the ŁOCK 0919 species from *Lactobacillus paracasei* to *Lactobacillus casei* [27,28,29].

The placebo group received maltodextrin—the medium in which the probiotic strains were suspended. The probiotic preparation and placebo were identical in appearance, packaging, and manner of administration. The probiotic preparation and placebo were supplied by Biomed S.A., Cracow, Poland, in a way that was suitable for blinded dispensing. In accordance with manufacturer’s recommendations, the products were stored at temperatures below 6 °C before and after being distributed to study doctors and dispensed to parents.

### 2.4. The Study Protocol

During the screening visit, medical history was taken from the subjects’ parents and physical examination was performed. Out of 286 children, 201 met the inclusion criteria. Having read the study protocol, the parents of 151 children provided their written informed consent to have their children participate in this clinical study. The parents of children who were included in the study were educated on how to maintain the elimination diet, store, and administer probiotics, and telephonically report any adverse effects or antibiotic therapy. On the first visit (baseline visit) investigators assessed AD severity with the SCORAD scale and allocated the children into either the probiotic or placebo group according to a computer-generated randomization list. Both the patients and investigators were blinded to group allocation. The probiotic preparation or placebo was administered orally after the contents of the sachet were dissolved in a small amount (approximately 10 mL) of water, once a day for three months. One month after the study intervention had been initiated, the parents brought their children to the next study visit, during with the children’s tolerance of the preparation was assessed and an amount of the probiotic or placebo sufficient for the next two months was dispensed. After the three-month intervention was completed, the children were examined by the investigator, and the severity of their disease was assessed with the SCORAD index. Nine months after the intervention was completed, the subjects were invited once again to a follow-up visit, during which AD symptoms were rated with the SCORAD index. Prior to the study intervention and nine months after its completion venous blood samples were collected from all subjects to assess the levels of total and allergen-specific IgE.

### 2.5. Endpoint Definitions

The primary outcomes included changes in AD symptom severity assessed with the SCORAD index and changes in the proportion of children with clinical improvement/no improvement or deterioration (symptom exacerbation).

The SCORAD index consists of the following components: A (20% of the final score) evaluates the extent of the lesions and is based on the rule of nines to express the percentage of the affected surface area on the body; B (60% of the final score) evaluates the intensity of six objective symptoms: erythema, edema/papules, scratch marks, oozing/crust formation, lichenification, and dryness, with each item graded on a scale from 0 to 3); and C (20% of the final score) evaluates subjective symptoms—itch and sleeplessness—both of which are graded on a 10-cm visual analog scale. The SCORAD index was calculated according to the following formula: A/5 + 7 B/2 + C. In this formula, A was defined as lesion extent (0–100), B was defined as objective symptom intensity (0–18), and C was defined as subjective symptom intensity (0–20). The maximum SCORAD score was 103.

All subjects were assessed for clinical improvement, no improvement, or exacerbation. A decline by >30% in the SCORAD index compared with baseline was considered to indicate a clinically significant improvement. A ≤30% decline in the SCORAD score was interpreted as no improvement. Finally, an increase in the SCORAD score in comparison to that at baseline was considered to indicate a clinical exacerbation.

The secondary study endpoints included the levels of total serum IgE and the presence of allergen-specific IgE.

The primary outcomes were assessed at three time points: at baseline, right after the three-month intervention, and after nine months of follow-up. The secondary outcomes were evaluated at baseline and after nine-month follow-up.

### 2.6. Specific and Total IgE

Allergen-specific IgE levels were measured with multiple allergen simultaneous tests (MAST)-immunoblot assays using Euroline Pediatric Profile (Euroimmun, AG, Lubeck, Germany), as described earlier by Konopka et al. [30]. A MAST-immunoblot assay can simultaneously detect allergen-specific IgE against 28 different allergens, including food allergens (egg white, egg yolk, cow’s milk, codfish, α-lactoalbumin, β-lactoglobulin, casein, bovine serum albumin, wheat flour, rice, soybean, peanut, hazelnut, carrot, potato, apple), inhalation allergens (grass mix, birch, mugwort), mites, fungi, molds, and animals (cat, dog, horse). A cross-reactive carbohydrate determinants marker was used as a control in each strip. Allergen-specific IgE levels of 0.35 kUA/L or greater were considered positive and indicated sensitization. Total IgE levels were measured using the ImmunoCap system according to manufacturer’s instructions.

### 2.7. Statistical Analyses

The collected data were analyzed using Stata Program version 12.1 by StataCorp LLC (College Station, TX, USA). The differences between the probiotic and placebo groups in terms of the sex and number of patients with clinical improvement/no improvement or exacerbation were evaluated with the use of Fisher’s exact test. The intergroup and intragroup differences in age, physical development parameters, SCORAD index data were evaluated with two-sided unpaired or paired *t*-tests after checking for equality of variances and normality using a Shapiro–Wilk test. If the normality assumption did not hold, two-sample Wilcoxon paired, or unpaired, signed-rank tests were used. The threshold of significance for all analyses was set at α = 0.05.

## 3. Results

### 3.1. Subjects

A total of 151 children were randomized to receive probiotics or placebo (Figure 1). During a three-month intervention period, a total of eight probiotic-group patients and six placebo-group patients were excluded from the study for the following reasons: antibiotic therapy (*n* = 4 in each group), refusal to take the probiotic (*n* = 2) or placebo preparation (*n* = 1), and failure to attend a study visit (*n* = 2 in the probiotic group and *n* = 1 in the placebo group). Additionally, parents of three children (*n* = 1 in the probiotic group and *n* = 2 in the placebo group) refused to continue the study due to AD symptom exacerbation. In consequence, a total of 134 children (66 receiving the probiotic preparation and 68 receiving placebo) completed the three-month intervention. Allergen sensitization was detected in 48 out of 66 (72.7%) and 48 out of 68 (70.6%) of subjects of the probiotic and placebo group, respectively (Table 1).

During the nine months of follow-up, 18 children from the probiotic group and 15 from the placebo group were withdrawn from the study because of no contact, change of residence, and failure to attend a visit. The study was completed by 48 patients supplemented with probiotics and 53 subjects receiving placebo. Allergen sensitized patients constituted 70.5% (*n* = 34) and 69.8% (*n* = 37) in the probiotic and placebo group, respectively.

Patient characteristics of those children who were included in the study and completed the three-month intervention have been presented in Table 1. Statistical analysis revealed no significant differences between the study groups in terms of these parameters. The mean age was similar in the two groups at 8.2 and 8.8 months in the probiotic and placebo group, respectively. Both groups were predominantly male, with males constituting 53.0% of the probiotic group and 70.6% of the placebo group. The vast majority of those included in the study had at least one parent or sibling with atopic disease (80.3% from the probiotic group and 76.5% from the placebo group). The proportion of breastfed subjects was low in both groups, at 15.2% and 16.2% in the probiotic and placebo group, respectively. The mean SCORAD score was slightly higher in the probiotic group (40.4 points) than in the placebo group (35.3 points). In both groups there were predominantly children with moderate AD (SCORAD score in the 25–50 point range); they constituted 51.5% of the probiotic group and 48.5% of the placebo group. In the probiotic group there were slightly more children with severe AD (SCORAD score of >50 points) than in the placebo group (27.3% and 17.6%, respectively).

The prevalence of specific antibodies is presented in Table 2. The vast majority of children had a multi-allergenic sensitization, 89.6% and 83.3% in the probiotic and placebo group, respectively. Most often children were sensitized to eggs, both egg white and yolk (more than 40% in each study groups). Specific IgE against CMP were found in 21.2% and 17.6% in the probiotic and placebo group, respectively. There were no statistical differences between the groups.

### 3.2. Changes in the SCORAD Score

The primary endpoints included changes in the SCORAD score in comparison with baseline. Table 3 presents the changes in SCORAD scores after the three-month treatment and after a nine-month follow-up. Both study groups showed a significant decrease in SCORAD scores in comparison with baseline, which indicates the effectiveness of elimination diet irrespective of probiotic intervention (Table 3). By the end of the three-month intervention the mean baseline SCORAD score in the probiotic and placebo groups decreased by 22.8 (*p* < 0.00001) and 16.7 (*p* < 0.00001) points, respectively; by the end of the nine-month follow-up the mean baseline score in the two groups decreased by 28.8 (*p* < 0.00001) and by 23.2 (*p* < 0.0001), respectively. Despite the SCORAD score decline being greater in the probiotic group, particularly in allergen sensitized patients, these differences were not statistically significant.

Interestingly, at the time of inclusion into the study, the mean SCORAD scores in both study groups were significantly higher in sensitized children than in those without IgE sensitization. The SCORAD scores for these two AD types were 45.1 and 28.0 (*p* = 0.001), respectively, in the probiotic group and 39.6 and 25.4 (*p* = 0.001), respectively, in the placebo group. After the three-month intervention, a statistically significant difference between the mean SCORAD scores between allergen sensitized patients and those with no sensitization was present only in the placebo group (22.0 versus 9.7; *p* = 0.004). Unlike the placebo group, the probiotic group showed similar mean SCORAD scores both in sensitized and not sensitized subjects (17.4 versus 18.3), which would indicate a considerable improvement (a decrease in the SCORAD score) following a three-month probiotic supplementation chiefly in allergen sensitized subjects. This was confirmed with an analysis of SCORAD score changes with respect to baseline. Only the probiotic group showed a significantly greater decrease in SCORAD scores in sensitized children in comparison with that in children with no allergen sensitization (by 27.8 and 9.7 points, respectively; *p* < 0.00001). These differences were no longer found after nine months of follow-up.

### 3.3. Improvement of AD Symptoms Assessed with the SCORAD Scale

Despite the fact that the probiotic and placebo groups did not differ significantly in terms of absolute SCORAD score changes, the probiotic group was significantly better in terms of the proportions of children who showed clinical improvement, no improvement, and deterioration after the three-month intervention (Table 4). Clinical improvement was defined as an over 30% decrease in the SCORAD score in comparison with the baseline score. At the end of the intervention, the probiotic group had a significantly higher percentage of children whose AD symptoms had improved (*p* = 0.029). The children receiving probiotic supplementation had a twofold higher chances of improving by the end of treatment than placebo-receiving children (odds ratio (OR) 2.56; 95% confidence interval (CI) 1.13–5.8; *p* = 0.012). A significant improvement following the probiotic intervention versus that in the placebo group was particularly noticeable in allergen sensitized children (*p* = 0.003). Probiotic supplementation improved the SCORAD score in over 90% of these children (44 out of 48 subjects included in the study); this was in contrast to less than 65% of children from the placebo group (31 out of 48 subjects included in the study) showing SCORAD-based improvement. The odds of achieving improvement in AD severity in probiotic-receiving sensitized children were six times greater than in the placebo group (OR 6.03; 95% CI 1.85–19.67; *p* = 0.001). AD severity exacerbation (increase in the SCORAD score) was observed in only one subject (2.1%) from this subgroup and in seven subjects (14.6%, *p* = 0.059) from the placebo group (OR 12; 95% CI 0.02–1.06; *p* = 0.028). Moreover, the number of children who exhibited no clinical improvement (a decrease in the SCORAD score of less than 30% of the baseline score) was lower in the probiotic group (*n* = 3, 6.2%) than in the placebo group (*n* = 10, 20.8%) (OR 0.25; 95% CI 0.07–0.99; *p* = 0.023).

Unlike in sensitized children, probiotic supplementation in those with no allergic sensitization failed to induce significant improvement compared with the level of disease severity observed in the placebo group (Table 4).

Follow-up assessments conducted nine months after treatment completion no longer showed the beneficial effects of probiotic supplementation (Table 5). Statistical analysis demonstrated no significant differences between the study groups. Although the proportion of sensitized children whose SCORAD scores improved from baseline was higher than that in the placebo group (91.2% and 78.4%, respectively), the difference did not reach statistical significance.

### 3.4. Secondary Endpoints

The mean level of total IgE at baseline was similar in the probiotic and placebo groups at 57.0 ± 98.0 kU/mL and 64.0 ± 95.4 kU/mL, respectively. After nine months of follow-up, both study groups showed increased total IgE levels, which were 189 ± 432.9 kU/mL and 177.6 ± 343.7 kU/mL in the probiotic and placebo group, respectively. The intergroup difference was not statistically significant.

Statistical analysis of allergen-specific IgE showed no significant differences between the study groups either at baseline or after nine months of follow-up (Table 2).

### 3.5. Tolerance of the Probiotic Preparation

Although no regular follow-up visits were planned to assess possible adverse effects, at each visit, the parents were asked about the child’s tolerance of the study preparation. Moreover, the parents had the option of calling an investigator to report any adverse effects. The study preparation was well tolerated, with only sporadic reports of adverse events (in both study groups) which most commonly involved changes in stool consistency. This occurred in three children from the probiotic group and in four from the placebo group.

## 4. Discussion

Probiotics are a potentially promising approach in the treatment of allergic conditions, including AD. The aim of the current multicenter, randomized, double-blind, placebo-controlled trial was to determine whether the probiotic preparation, containing a mixture of *Lactobacillus rhamnosus* ŁOCK 0900, *Lactobacillus rhamnosus* ŁOCK 0918, and *Lactobacillus casei* ŁOCK 0919, would be effective in children under two years old with AD and CMP allergy. The results of our study showed that the probiotic preparation is superior to placebo in terms of primary outcomes, and this was largely due to the observed benefit in improving symptom severity in allergen sensitized patients. The results of this multicenter clinical trial are consistent with our earlier “preliminary” data published in Polish-language journals [31,32].

Literature reports on probiotic supplementation in AD and food allergy are conflicting. A systematic review and meta-analysis of 39 randomized controlled trials involving 2599 participants conducted by Makrgeorgou et al. in 2018 showed that the currently available probiotic strains probably make little or no difference in improving AD symptoms compared with placebo [33]. However, this study included participants aged from one year to 55 years (with six of the analyzed studies conducted in adults only). Therefore, the age range in this meta-analysis was relatively wide, which may affect the conclusions. In contrast to the meta-analysis by Makrgeorgou et al., a recently published (2020), updated meta-analysis by Jiang et al. demonstrated that interventions with probiotics potentially lower the incidence of AD and relieve AD symptoms in children, particularly when treating children aged over 1 year [34]. Studies included in this meta-analysis with participants aged under one year reported no significant results [34]. Zhao et al., whose systematic review and meta-analysis focused on the effectiveness of probiotics in AD infants (defined as subjects under the age of 36 months) presented that probiotic treatment was beneficial compared to placebo in this age group [35]. This meta-analysis included eight randomized placebo-controlled studies (741 subjects) and revealed that probiotic preparations containing *Lactobacillus* species had a protective effect in infants with moderate-to-severe AD. Therefore, the age of study subjects may be a factor that affects the clinical effectiveness of probiotics.

The subjects included in our study were under two years old, with the mean age in the two study groups under nine months. This age range was selected based on an assumption that probiotics are the most effective during early development, when the composition of intestinal microbiota is being established (this process is typically completed by the age of 2–3 years) and the immune system is being programmed for the future [9]. Our earlier studies demonstrated that ŁOCK strain supplementation in infants with AD and food allergy modifies the composition of their intestinal microbiota [36]. The group that received a mixture of ŁOCK strains 0900, 0908, and 0919 showed a significantly higher proportion of subjects with an abundance of *Bacteroidetes* [36]. Although the microbiome analysis was not performed in the present study, we believe that one of the effects of ŁOCK strain supplementation is a modification of the gut microbiota in young children.

It is not only the age at which probiotics are administered that affects their effectiveness. The dose and duration of treatment are equally important. In our study, probiotics were administered at 10^9^ CFU once daily for three months. The beneficial effect was observed after the end of the probiotic intervention and was not extended for the further nine months of follow-up. The treatment period in most of the clinical studies evaluating the efficacy of probiotics in AD lasted no more than three months [11,12,13,14,37,38], but the intervention time was extended up to six months in AD prevention studies [19,20,21,22]. A meta-analysis by Jiang et al. showed that the SCORAD scores in studies with treatment periods of >8 weeks decreased more than in studies with treatment periods of <8 weeks [34], but the authors did not assess the optimal duration of probiotic intervention. It is therefore possible that an intervention lasting more than three months would produce longer-lasting effects. Interestingly, the decreases in mean SCORAD scores from baseline that were observed in our study after one year were significant in both study groups (reaching 12.5 points in the probiotic group and 13.1 points in the placebo group), which indicates that the elimination diet followed by all subjects was highly effective.

AD, particularly the extrinsic (with increased IgE levels and specific sensitization) subtype is often the first step in development of other atopic manifestations in allergic march [39]. Carlsten et al. presented that the early onset of AD (i.e., AD during the first two years of life) was associated with food allergy (OR 13.4; 95% CI 2.94–61.4), allergic rhinitis (OR 3.47; 95% CI 1.34–8.99), and asthma (OR 7.48; 95% CI 2.53–22.2) [40]. Hulst et al.’s analysis of 13 prospective studies showed that approximately 30% of young children with AD will develop asthma by the age of six years [41]. It is considered that allergic morbidity profiles is dependent on IgE sensitization [42]. Gabet et al. demonstrated that sensitization to egg white and CMP at age 18 months significantly increased the risk of asthma later in childhood [43]. Thus, the early onset of AD may be a window of opportunity to modify the profile sensitization by therapeutic interventions, including probiotics [12,13]. Elazab et al. performed a meta-analysis of randomized, placebo-controlled trials to assess the effects of probiotic supplementation on atopic sensitization and asthma/wheeze prevention in children [44]. They presented that probiotics were effective in reducing total IgE and the risk of atopic sensitization when administered prenatally and postnatally, but probiotic intervention did not significantly reduce asthma/wheeze occurrence.

Our study showed that allergen sensitization was associated with a more severe AD, and the probiotic intervention with ŁOCK strains was more effective in sensitized patients compared to those without IgE sensitization. Although we attempted to evaluate the effect of probiotics on the sensitization profile after nine months of follow-up, we did not demonstrate the effectiveness of probiotic intervention on total IgE levels or on allergen sensitization. Other authors have also observed that probiotics are superior in sensitized subjects with AD [11,37,45], but most of them did not follow up after the intervention and did not analyze the effect of probiotics on the sensitization profile. Long-term effects of a probiotic intervention in infants with AD were demonstrated by van der Aa et al. in their study involving administration of an extensively hydrolyzed formula with *Bifidobacterium breve* M-16V and a galacto/fructooligosaccharide mixture to infants with AD for a period of 12 weeks [45]. The authors did not find any difference in SCORAD indices between the probiotic and the placebo group, but in the subgroup of sensitized infants, the SCORAD improvement at week 12 was significantly greater in the probiotic than in the placebo group [45]. After one year, van der Aa et al. observed that the prevalence of “frequent wheezing” and “wheezing and/or noisy breathing apart from colds” was significantly lower in the probiotic than in the placebo group (13.9% versus 34.2%) [46]. The total IgE levels did not differ between the two groups, but only children in the placebo group (15.2%) developed elevated IgE antibodies against cat. Canani et al. reported that administration of extensively hydrolyzed casein formula containing LGG reduced the occurrence of other allergic manifestations in infants with a food allergy to CMP [47]. That randomized placebo-controlled study had a three-year follow-up period. Administration of probiotic-enriched casein hydrolysates not only stopped the allergic march but also activated the development of oral tolerance.

Thus, it seems that early probiotic intervention may have impact on the sensitization profile and on developing more severe allergies, such as asthma. Further studies in a much larger group of children are needed to confirm this conclusion in term of strains evaluated in our study.

### 4.1. The Mechanism of Action of ŁOCK Strains

Experimental studies in germ-free mice associated with the mixture of *Lactobacillus rhamnosus* ŁOCK 0900, *Lactobacillus rhamnosus* ŁOCK 0908, and *Lactobacillus casei* ŁOCK 0918 demonstrated that gut colonization with ŁOCK strains affects the formation and maturation of the epithelial gut barrier, mostly via activation of proteins (zonulin, and occludin) that play a role in the formation of junctions between intestinal epithelial cells [48]. The reported effects of ŁOCK strains may be of great importance in children with food allergies, who exhibit increased permeability of the intestinal epithelial barrier. Moreover, colonization with these strains was shown to activate secretory IgA production in the gut, which additionally strengthens the intestinal barrier and enhances protection against infectious and toxic agents, including allergens. An experimental model of allergy to birch pollen (Bet v1) demonstrated that gut colonization suppressed Bet v1 sensitization and lowered total IgE levels, which was associated with regulatory T-cell activation and immune tolerance development [48].

Cultures of peripheral blood cells obtained from children with AD showed that a ŁOCK strain mixture suppresses the pro-allergic Th2 cytokine profile and stimulates the production of Th1-derived cytokines and transforming growth factor beta (TGF-β)—a factor responsible for immune tolerance development [24]. Apart from the described effect on strengthening the gut epithelial barrier, nonspecific immunity, and activating regulatory T-cells (which play an important role in the maintaining a balance between the pro-allergic and pro-inflammatory cytokine profiles, and in the development of immune tolerance), ŁOCK strains may modify the composition of gut microbiota [48]. In vitro studies showed that these strains are characterized by their high antagonism to many pathogens, including *Staphylococcus aureus* [23].

### 4.2. Limitations and Strengths of the Study

One of the strengths of this study is its design as a multicenter randomized double-blind placebo controlled trial. The study group was homogeneous, as it consisted of children with AD (diagnosed according to Hanifin’s and Rajka’s criteria) and a concomitant food allergy to CMP (confirmed via an open elimination–provocation test). Another strength of the study was the fact that AD severity was assessed with the SCORAD index, although the subjective nature of this index may be considered somewhat of a limitation as well. We suspect that assessments conducted by two specialists (in our study it was only one allergy specialist or a dermatologist) would be more objective. The effect of probiotic intervention on the composition of the gut microbiota was not analyzed in this study, which should also be considered a significant limitation. In addition, evaluation of the presence of the administrated strains in feces during the intervention and in the 9-month follow up would show the ability of the ŁOCK strains to survive and multiply in the intestines. Another limitation of our study is associated with the lack of systematic verification of the actual probiotic/placebo administration or checking for the use of other dietary supplements.

## 5. Conclusions

This multicenter randomized, double-blind, placebo-controlled study in children up to two years old with AD and CMP allergy shows that administration of probiotic preparation containing a mixture of *Lactobacillus rhamnosus* ŁOCK 0900, *Lactobacillus rhamnosus* ŁOCK 0908, and *Lactobacillus casei* ŁOCK 0918 strains is safe and induces beneficial effects especially in allergen sensitized patients. Supplementation of the children’s diet with the probiotic preparation for three months resulted in a significant improvement in AD symptom severity assessed with the use of the SCORAD index.

## Figures and Tables

**Figure 1 nutrients-13-01169-f001:**
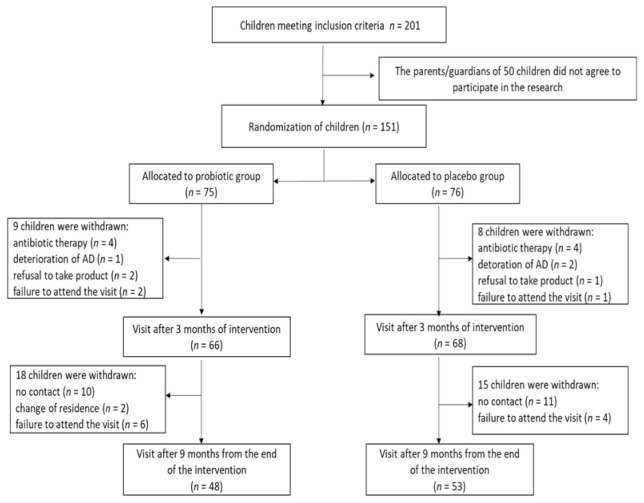
Study protocol flowchart.

**Table 1 nutrients-13-01169-t001:** Patient characteristics.

	Probiotic Group *(n* = 66)	Placebo Group (*n* = 68)
Age in months (range)	8.2 ± 6.1(4–23)	8.8 ± 6.6 (2–23)
Sex—*n* (%)		
Male	37 (56.1)	48 (70.6)
Female	29 (43.9)	29 (29.4)
Weight in kg (range)	9.55 ± 3.6 (6.7–14.2)	9.01 ± 3.1 (6.1–13.9)
Length/height in cm (range)	76.95 ± 11.3 (65–89)	75.8 ± 12.5 (57–92)
Family positive history for atopy—*n* (%)	53 (80.3)	52 (76.5)
Breastfeeding—*n* (%)	10 (15.2)	11 (16.2)
SCORAD score in points (range)	40.4 ± 20.0 (14–103)	35.3 ± 17.7 (12–99)
AD severity (SCORAD)		
Mild (score <25)—*n* (%)	14 (21.2)	23 (33.3)
Moderate (score 25–50)—*n* (%)	34 (51.5)	33 (48.5)
Severe (score >50)—*n* (%)	18 (27.3)	12 (17.6)
AD type		
Allergen sensitization—*n* (%)	48 (72.7)	48 (70.6)
No allergen sensitization—*n* (%)	18 (27.3)	20 (29.4)

The table includes the data of only those children who completed the three-month intervention. The results have been presented as means ± standard deviation, with the minimum and maximum values, or range, in parentheses; or the number of subjects, with the percentage in parentheses. Atopic dermatitis (AD) severity was assessed with use of the SCORing Atopic Dermatitis (SCORAD) index.

**Table 2 nutrients-13-01169-t002:** The prevalence of specific IgE antibodies.

Allergen	Baseline		9-Month Follow-Up	
	Probiotic Group (*n* = 66) Number (%)	Placebo Group (*n* = 68) Number (%)	Probiotic Group (*n* = 48) Number (%)	Placebo Group (*n* = 53) Number (%)
Food allergens				
Egg white	30 (45.4)	29 (42.6)	20 (41.2)	23 (43.4)
Egg yellow	23 (34.8)	20 (29.4)	15 (31.2)	15 (28.3)
Hazelnut	15 (22.7)	13 (19.1)	15 (31.2)	13 (24.5)
Cow’s milk	14 (21.2)	12 (17.6)	7 (14.6)	7 (13.2)
α-lactoalbumin	9 (13.6)	8 (11.8)	4 (8.3)	6 (11.3)
β-lactoglobulin	6 (9.1)	5 (7.3)	3 (6.2)	5 (9.4)
casein	7 (10.6)	6 (8.8)	4 (8.3)	5 (9.4)
Potato	11 (16.7)	10 (14.7)	8 (16.7)	8 (15.1)
Wheat flour	8 (12.1)	10 (14.7)	8 (16.7)	7 (13.2)
Codfish	8 (12.1)	7 (10.3)	6 (12.5)	7 (13.2)
Soybean	7 (10.6)	5 (7.3)	7 (14.6)	9 (17.0)
Peanut	4 (6.1)	3 (4.4)	7 (14.6)	7 (13.2)
Apple	2 (3.0)	3 (4.4)	7 (14.6)	7 (13.2)
Carrot	1 (1.5)	2 (2.9)	2 (4.2)	2 (3.8)
Rice	1 (1.5)	0	1	0
Other allergens				
Mites	5 (7.6)	6 (8.8)	7 (14.6)	6 (11.3)
Grass mix	6 (9.1)	5 (7.3)	6 (12.5)	5 (9.4)
Birch	5 (7.6)	5 (7.3)	8 (16.7)	5 (9.4)
Cat	4 (6.1)	4 (5.9)	6 (12.5)	5 (9.4)
Dog	4 (6.1)	3 (4.4)	6 (12.5)	5 (9.4)
Hourse	2 (3.0)	1 (1.5)	2 (4.2)	1 (1.9)
*Alteria alternate*	0	1 (1.5)	2 (4.2)	2 (3.8)
Mugwort	0	1 (1.5)	0	0

The table includes the data at baseline and after 9 months of follow up. The results are presented as a number of children (percentage in parentheses) sensitized to specific allergen. The specific IgE antibodies were detected using MAST-immunoblot. Patients presented allergen-specific IgE ≥ 0.35 kUA/L were considered as sensitized.

**Table 3 nutrients-13-01169-t003:** The effect of probiotic intervention on changes in SCORAD scores in infants with AD.

Groups	Baseline	after 3-Months Intervention				after 9-Months Follow Up			
	Mean ± SD	Mean ± SD	Change from Baseline	*p*-Value Within-Group	*p*-Value Comparison with Placebo	Mean ± SD	Change from Baseline	*p*-Value Within-Group	*p*-Value Comparison with Placebo
All patients with AD									
Probiotic Placebo	40.4 ± 20.0 35.7 ± 17.8	17.6 ± 14.8 18.9 ± 17.5	−22.8 ± 17.5 −16.7 ± 17.9	<0.00001 <0.00001	0.881	12.5 ± 15.4 13.1 ± 13.3	−28.8 ± 17.1 −23.2 ± 20.9	<0.00001 <0.00001	0.704
Patients with allergen sensitization									
Probiotic Placebo	45.1 ± 20.1 * 39.6 ± 17.5 ^#^	17.4 ± 15.9 22.0 ± 18.1	−27.8 ± 16.8 ** −17.5 ± 18.0	<0.00001 <0.00001	0.186	13.2 ± 17.3 14.7 ± 13.4	−31.2 ± 20.8 −24.5 ± 18.6	<0.00001 <0.00001	0.289
Patients without sensitization									
Probiotic Placebo	28.0 ± 13.8 * 25.4 ± 14.4 ^#^	15.3 ± 11.6 11.7 ± 12.9	−11.7 ± 11.6 ** −13.5 ± 17.9	0.004 0.002	0.109	10.8 ± 9.6 8.0± 12.1	−17.4 ± 19.7 −16.6 ± 16.7	0.013 0.003	0.486

The results are presented as arithmetical mean ± standard deviation. Intra- and inter-group differences were calculated using a paired t-test after checking the normal distribution. In addition, statistical analyses were done between sensitized and not sensitized subgroups in the same study group with the use of non-paired tests. *p*-values < 0.05 * statistically significant differences in sensitized patients versus patients with no sensitization in the probiotic group, *p* = 0.001; ** *p* < 0.00001; # statistically significant differences in sensitized patients versus patients with no sensitization in the placebo group, # *p* = 0.001.

**Table 4 nutrients-13-01169-t004:** The effect of the mixture of probiotic *Lactobacillus* ŁOCK strains on AD symptom improvement or exacerbation after 3 months of intervention.

Groups	Improvement	Deterioration	No Improvement
All patients with AD			
Probiotic group (*n* = 66)	55 (83.3)	4 (6.1)	7 (10.6)
Placebo group (*n* = 68)	45 (66.2)	9 (13.2)	14 (20.6)
*p*-value	0.029	0.128	0.154
OR (95% CI)	2.56 (1.13–5.80)	0.42 (0.12–1.45)	0.46 (0.17–1.22)
*p*-value for OR	0.012	0.171	0.119
Patients with allergen sensitization			
Probiotic group (*n* = 48)	44 (91.7)	1 (2.1)	3 (6.2)
Placebo group (*n* = 48)	31 (64.6)	7 (14.6)	10 (20.8)
*p*-value	0.003	0.059	0.070
OR (95% CI)	6.03 (1.85–19.67)	0.12 (0.02–1.06)	0.25 (0.07–0.99)
*p*-value for OR	0.001	0.028	0.023
Patients without sensitization			
Probiotic group (*n* = 18)	11 (61.1)	3 (16.7)	4 (22.2)
Placebo group (*n*= 20)	14 (70.0)	2 (10.0)	4 (20.0)
*p*-value	0.503	0.209	0.101
OR (95% CI) *p*-value for OR	0.79 (0.21–2.92) 0.359	1.8 (0.27–12.2) 0.547	1.14 (0.24–5.44) 0.433

The table shows the number of subjects (percentage in parentheses) who showed clinical improvement, deterioration (i.e., symptom exacerbation), or no improvement based on SCORAD scores in comparison with baseline scores. Clinical improvement was defined as an over 30% decrease in the SCORAD score from baseline; no improvement was defined as a less than 30% decrease in the SCORAD score; symptom AD exacerbation was defined as no decrease in the SCORAD score from baseline. Statistical analysis used Fisher’s exact test. OR = odds ratio, CI = confidence interval.

**Table 5 nutrients-13-01169-t005:** The effect of the mixture of probiotic *Lactobacillus* strains on AD symptom severity after 9 months of follow-up.

Groups	Improvement	Deterioration	No improvement
All patients with AD			
Probiotic group (*n* = 48)	41 (85.4)	7 (14.6)	0
Placebo group (*n* = 53)	42 (79.2)	6 (13.3)	5 (9.4)
*p*-value	0.448	0.769	0.058
OR (95% CI)	1.53 (0.54–4.34)	1.33 (0.42–4.30)	0.09 (0.005–1.69)
*p*-value for OR	0.420	0.626	0.108
Patients with allergen sensitization			
Probiotic group (*n* = 34)	31 (91.2)	3 (8.8)	0
Placebo group (*n* = 37)	29 (78.4)	3 (8.1)	5 (13.5)
*p*-value	0.675	1.0	0.055
OR (95% CI)	2.85 (0.69–11.79)	1.10 (0.21–5.84)	0.09 (0.005–1.61)
*p*-value for OR	0.148	0.457	0.101
Patients without sensitization			
Probiotic group (*n* = 14)	10 (71.4)	4 (28.6)	0
Placebo group (*n* = 16)	13 (81.3)	3 (18.7)	0
*p*-value	0.675	0.682	1.000
OR (95% CI) *p*-value for OR	0.58 (0.10–3.19) 0.528	1.73 (0.31–9.57) 0.528	1.13 (0.002–61.08) 0.949

The table shows the number of subjects (percentage in parentheses) who showed clinical improvement, deterioration (i.e., symptom exacerbation), or no improvement based on SCORAD scores in comparison with baseline scores. Clinical improvement was defined as an over 30% decrease in the SCORAD score from baseline; no improvement was defined as a less than 30% decrease in the SCORAD score; symptom AD exacerbation was defined as no decrease in the SCORAD score from baseline. Statistical analysis used Fisher’s exact test. OR = odds ratio, CI = confidence interval.

## Data Availability

The data presented in this study are available on request from the corresponding author.

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
