# Peer review of "The Effectiveness of Probiotic Lactobacillus rhamnosus and Lactobacillus casei Strains in Children with Atopic Dermatitis and Cow’s Milk Protein Allergy: A Multicenter, Randomized, Double Blind, Placebo Controlled Study"

_nutrients, 2021, doi:10.3390/nu13041169_

Round 1
Reviewer 1 Report
A well-structured study on the effect of probiotic Lactobacillus rhamnosus and Lactobacillus casei strains in children with atopic dermatitis and food allergy. I have some queries:
In the materials and methods section, please state the maker of the statistical program and its city and country.
Why did you prefer to use Fisher exact test instead of a chi-square test?
In the introduction, a small paragraph on current and emerging biological treatment of AD is in my opinion a great addition to the study; here's an article you could consider: doi: 10.1111/dth.12787.
Thank You
Author Response
All authors would like to thank you for revision of our manuscript and all comments and suggestion which allow us to improve the manuscript. Changes made in the manuscript are marked in red (added in the track changes mode). Below are our answers to all comments:
- In the materials and methods section, please state the maker of the statistical program and its city and country.
In the revised version of the manuscript, this information has been completed (line 233-234)
- Why did you prefer to use Fisher exact test instead of a chi-square test? Did the authors excluded also parasitic or bacterial intestinal infestation/infection?
Generally, the Fisher’s exact test is preferable to the chi-squared test because it is an exact test (accuracy of the Fisher’s exact test is exact, whereas chi-squared is approximate). The chi-squared is for large amount of observation, and this test should be particularly avoided if there are few observations (e.g. less than 10) for individual cells. In our study the number of observation was low (e.g. 1 versus 7; 4 versus 9, Table 4, line 366; or 0 versus 5; 3 versus 3, Table 5, line 373). In contrast, the Fisher’s exact test is dedicated to smaller amounts of observation, and to table contingency2x2. That is why we preferred the Fisher’s exact test.
- In the introduction, a small paragraph on current and emerging biological treatment of AD is in my opinion a great addition to the study; here's an article you could consider: doi: 10.1111/dth.12787.
Thank you for this suggestion. We added such information in corrected version of the manuscript in the Introduction section with the suggested citation - lines 48-53.
Lines 48-52:
….”AD is characterized by epithelial barrier defects and dysregulation of both the innate and adaptive immune response against different triggers. Recently, cytokines and other mediators that play an important role in the pathogenesis of skin inflammation have become a target for new forms of therapy [2,3]. Drugs for which interleukin (IL)-4 and IL-13 – the main cytokines of T helper 2 (Th2) immune activation – are the targets, are particularly represented. Dupilumab, a human anti-IL-4 receptor α monoclonal antibody that blocks IL-4- and IL-13-mediated signaling pathways, is the first biological drug approved by Food and Drug Administration for the treatment of moderate-to-severe AD in adolescents and adults [3]. Another approved therapeutic option for topical use is crisaborole, a phosphodiesterase-4 inhibitor which is a key regulator in the inflammatory cytokine cascade [3]”….

Reviewer 2 Report
Thank you for this very interesting paper. I think this is an important addition to the literature.
I have only one comment: In the abstract (line 40) and elsewhere you mention "IgE-mediated AD". I do not think this is an accepted concept as you have written it. I think a more conventional term would be "extrinsic AD". But I realize that you are saying something beyond this. Would you consider saying "IgE-triggered AD" or "food allergy-associated AD"? Especially because it was not a double blind placebo controlled food challenge, I just don't know if we can conclude that the AD was truly IgE-mediated.
Reference: Czarnowicki T, He H, Krueger JG, Guttman-Yassky E. Atopic dermatitis endotypes and implications for targeted therapeutics. Journal of Allergy and Clinical Immunology. 2019 Jan 1;143(1):1-1.
Author Response
All authors would like to thank you for revision of our manuscript and all comments which allow us to improve the manuscript. Changes made in the manuscript are marked in red. Below is the answer to your comment:
- In the abstract (line 40) and elsewhere you mention "IgE-mediated AD". I do not think this is an accepted concept as you have written it. I think a more conventional term would be "extrinsic AD". But I realize that you are saying something beyond this. Would you consider saying "IgE-triggered AD" or "food allergy-associated AD"? Especially because it was not a double blind placebo controlled food challenge, I just don't know if we can conclude that the AD was truly IgE-mediated.
Thank you for this suggestion. Your comment and reading the proposed literature has led to a change in the terminology describing the subgroup of patients with AD and positive IgE antibodies into “the allergen sensitized group”. The novel terminology has been used throughout the corrected manuscript (marked in red). Additionally, in the Discussion, we added a paragraph regarding the novel term used and we cited the necessary publications emphasizing the importance of such a decision (line 443-444, and 449-466). New references have also been completed in the revised manuscript (number 39, 43, 44).

Reviewer 3 Report
I read with interest this manuscript aiming at assessing the efficacy of specific strains of probiotics in children with atopic dermatitis and cow’s milk allergy. This multi-centered, randomized, double blind, placebo controlled study randomized patients to intervention versus placebo for 3 months and determined the effect on atopic dermatitis symptom severity over a year period.
The article deals with an interesting topic. The manuscript is clear but English language is not very well written and requires spell check. I would suggest to modify the title and substitute “food allergy” with “cow’s milk allergy”. Length of the introduction is good respect to the length of the manuscript and the conclusions respond to the aim of the study.
In the introduction or discussion, when reporting effects on primary prevention of allergies, I suggest to report the recommendations of the World Allergy Organization: GLAD-P probiotics document.
2.2 Patients
Lines 110-113 Inclusion and exclusion criteria are the same regarding age of enrolment.
3.1 Subjects
Line 237 why patients with AD symptoms exacerbation have been excluded?
Line 242-245 I’m not sure that the nomenclature IgE-mediated AD and non-IgE-mediated AD is correct. I suggest to use the term sensitization.
Discussion
Lines 409-411 it is an incomplete sentence.
Lines 424-428 It would have been interesting to analyse gut microbiota in enrolled children at the different time points for its composition and persistence in its different strains. This should be reported in the limitations.
Figures and tables are necessary and self-explaining but I would suggest for table 2 to specify at what time point the reported specific IgEs refer to. Findings of the other 2 time points should be reported and discussed. I was wondering why the number of patients tested became 48 in each group and not 48 in intervention group and 53 in the placebo group, is it a typo? In table 1 positive “family” history for atopy must be specified.
Only few references are recent, please check for more recent papers.
Author Response
All authors would like to thank you for revision of our manuscript and all comments and suggestions which allow us to improve the manuscript. Changes made in the manuscript are marked in red (added in the track changes mode). Below are our answers to all comments:
- The manuscript is clear but English language is not very well written and requires spell check. I would suggest to modify the title and substitute “food allergy” with “cow’s milk allergy”.
According to the Reviewer’s suggestion the language of the manuscript was checked, and the title was modified – the term “food allergy” in the title was changed into “cow’s milk protein allergy (line 4)
- In the introduction or discussion, when reporting effects on primary prevention of allergies, I suggest to report the recommendations of the World Allergy Organization: GLAD-P probiotics document.
Thank you for this suggestion. The paper with the recommendation of WAO was added into the references (the number 13), and the recommendations were presented in the Introduction section (line 77-79) and reported in the discussion section.
- Lines 110-113 Inclusion and exclusion criteria are the same regarding age of enrolment.
It was mistake, and it has been corrected (line 123).
- Line 237 why patients with AD symptoms exacerbation have been excluded?
Thank you for this comment. Patients were not excluded by researchers, but parents of the children with symptom exacerbation did not agree to continue the study. This is clearly explained in the revised version of the manuscript (line 250-252).
- Line 242-245 I’m not sure that the nomenclature IgE-mediated AD and non-IgE-mediated AD is correct. I suggest to use the term sensitization
Thank you for this suggestion. We changed the term IgE-mediated AD into “sensitization”. The term “IgE-mediated” has been changed throughout the revised version of the manuscript (changes are marked in red).
- Lines 409-411 it is an incomplete sentence.
Yes, You are right. The sentence has been completed (lines 409-412).
“ Zhao et al., whose systematic review and meta-analysis focused on the effectiveness of probiotics in AD infants (defined as subjects under the age of 36 months) presented that probiotic treatment was beneficial compared to placebo in this age group [35].”
- Lines 424-428 It would have been interesting to analyse gut microbiota in enrolled children at the different time points for its composition and persistence in its different strains. This should be reported in the limitations.
Unfortunately, we did not analyze the gut microbiome in this study, and this is emphasized in the revised version of the manuscript in discussion section and as significant limitations of the study (lines .425-427, and lines 519-524).
- Figures and tables are necessary and self-explaining but I would suggest for table 2 to specify at what time point the reported specific IgEs refer to. Findings of the other 2 time points should be reported and discussed. I was wondering why the number of patients tested became 48 in each group and not 48 in intervention group and 53 in the placebo group, is it a typo? In table 1 positive
Thank you for this part of the review. In table 2 (in the first version of the manuscript) we presented IgE sensitization only in one time point – at baseline, and the numbers 48 in the probiotic group, and 48 in the placebo group were correct as they only referred to sensitized patients (not to all patients in the group). In revised version of the manuscript, we changed the Table 2 (line 289), and all patients in the groups were used for % calculations. Currently, Table 2 contains information on the 2 time points: baseline and after 9 months of follow-up (the blood for specific IgE was not taken after 3 months, and this information is given in the section “The study protocol, lines 192-194, and in section “Endpoint definitions”, lines 217-219). The results on IgE sensitization are presented in section “Results” (lines 352-359), and are discussed in “Discussion” sections (lines 449-466).
- “Family” history for atopy must be specified
This information was added (Table 1, line 284).
- Only few references are recent, please check for more recent papers
According to the suggestion we added more recent references in revised version of the manuscript (numbers of references 1, 2, 3, 5, 12, 37, 38. 39, 43).

Round 2
Reviewer 1 Report
The authors responded to all queries. The paper is publishable